

# Morpho-physiological traits and yield quality for cassava genotypes planted under drought during canopy establishment

Passamon Ittipong[1], Poramate Banterng[1,2], Nimitr Vorasoot[1], Sanun Jogloy[1,2], Piyada Theerakulpisut[3], Kochaphan Vongcharoen[4] and Supranee Santanoo[3]

[1] Faculty of Agriculture, Khon Kaen University, Khon Kaen, Thailand
[2] Plant Breeding Research Center for Sustainable Agriculture, Khon Kaen University, Khon Kaen, Thailand
[3] Faculty of Science, Khon Kaen University, Khon Kaen, Thailand
[4] Faculty of Science and Health Technology, Kalasin University, Kalasin, Thailand

Corresponding author
Poramate Banterng,
bporam@kku.ac.th

## ABSTRACT

Growth analysis provides better insight into the adaptability of cassava genotypes grown under drought conditions during canopy establishment and full irrigation. This study is intended to determine the growth rate and starch yield of different cassava genotypes grown under irrigation and drought treatments during canopy establishment. The experiment was conducted in two growing seasons at Khon Kaen University, Thailand, from August 2021 to August 2022 (2021/2022) and from August 2022 to August 2023 (2022/2023) using six cassava genotypes. A $2 \times 6$ split-plot design with four replications was used. The main plots were full irrigation and drought conditions during canopy establishment (90 to 150 days after planting (DAP)), and the cassava genotypes were assigned as subplots. The results showed that drought treatment applied during 90 to 150 DAP reduced relative water content (RWC) in cassava leaves for both growing seasons, storage root growth rate (SRGR) in the 2021/2022 growing season, and stem growth rate (SGR) and crop growth rate (CGR) in the 2022/2023 growing season. Re-watering after a drought supported cassava's growth rate, resulting in desirable yield and biomass at final harvest. The Rayong 72 and CMR38-125-77 produced significantly higher storage root dry weight, harvest index (HI), and starch yield than the other tested genotypes. Growing under drought treatment, the best performance in storage root dry weight with statistical significance for both years was recorded for CMR38-125-77 (11.2 and 11.4 t ha$^{-1}$ for the 2021/2022 and 2022/2023 growing seasons, respectively), and this was associated with a high crop growth rate (CGR, 12.3 g m$^{-2}$ day$^{-1}$ for the 2021/2022 growing season) and relative growth rate (RGR, 1.11 $\times$ 10$^{-2}$ g g$^{-1}$ day$^{-1}$ for the 2022/2023 growing season) during 180 to 360 DAP. These favorable cassava genotypes should be utilized for future plant breeding programs and cultivation to achieve the desired productivity in the growing areas with drought during canopy establishment.

## INTRODUCTION

Cassava (*Manihot esculenta* Crantz) is extensively cultivated in Africa, Asia, and Latin America, and it plays a vital role in food, animal feed, and bioethanol production (*Bayata, 2019*; *Ferguson et al., 2019*). In 2022, Thailand was a major cassava producer, with an output of 35.10 million tons, the harvested area was 1.67 million hectares, and the average yield was 21.44 tons per hectare (*Office of Agricultural Economics, 2023*). However, the average yield in Thailand was lower than the yield potential of 75 tons per hectare (*Boonseng, 2011*; *Kongsil et al., 2024*). The major cassava growing area in Thailand is in the Northeast, characterized by sandy soils with poor soil fertility, low soil water holding capacity, and unpredictable rainfall. Cassava in this region is typically cultivated in two seasons: the main rainy and the late rainy seasons (*Polthanee, 2018*). For growing cassava in the late rainy season, storage root yield can be affected by drought during the early growth phase, specifically the canopy establishment, and it causes a decrease in yield by approximately 32 to 60 percent (*Palanivel & Shah, 2021*). There are several options for increasing cassava yields in drought-prone areas. These include the application of supplemental irrigation and the selection of suitable cassava genotypes. Recommending drought-adaptive cassava genotypes is a strategy to help farmers achieve high productivity with low investment.

Determinations of the agronomic traits, physiological traits, and starch content of cassava genotypes have been done for different water regimes. Photosynthesis, growth, productivity, and nutrient use efficiency among cassava genotypes under rain-fed conditions were documented by *El-Sharkawy & De Tafur (2010)*. In arid and semi-arid lands, different cassava genotypes were evaluated under drought and irrigated conditions in agro-climatic zone five (ACZ-V) (*Orek et al., 2020*). *Wongnoi et al. (2020)* studied the performance of different cassava genotypes in upland in a dry environment during the high storage root accumulation stage. Various cassava genotypes grown under different irrigation levels (100%, 60%, and 20% crop water requirement (ET crop)) during the early growth phase were reported (*Ruangyos et al., 2024*). *Mahakosee et al. (2019)* reported a Rayong 9 cassava genotype grown under rain-fed and irrigated conditions. Growth and yield of cassava genotypes grown under rain-fed upper paddy field conditions were assessed (*Sawatraksa et al., 2018*; *Sawatraksa et al., 2019*). These studies did not cover the performances of some cassava genotypes for drought conditions during the canopy establishment and under full irrigation.

Photosynthesis, carbohydrate partitioning, growth, and yield were studied in different cassava genotypes under full irrigation and early drought conditions (*Santanoo et al., 2024*). However, this report was only based on a single experiment, necessitating further research for more robust conclusions. In addition, morpho-physiological traits and yield quality based on growth analysis for cassava offer valuable insights into crop growth habits, aiding in the selection of suitable cassava varieties for various environments (*Phuntupan & Banterng, 2017*; *Phoncharoen et al., 2019a*; *Sawatraksa et al., 2019*; *Ruangyos et al., 2024*). The information on growth analysis can help design suitable cassava genotypes for the dry period during the early growth phase and provide appropriate water management practices. Growth analysis for cassava on the basis of crop growth rate (CGR), stem growth rate (SGR),

leaf growth rate (LGR), storage root growth rate (SRGR), and relative growth rate (RGR) for different cassava genotypes can support a better understanding of cassava adaptability in different growing environments. Previous studies mentioned growth analysis for cassava growing under different nitrogen fertilizer applications (*Phuntupan & Banterng, 2017*), various environments (*Sawatraksa et al., 2019*), and different planting dates (*Phoncharoen et al., 2019a*). However, an investigation on the performance of different cassava genotypes in terms of growth rate under non-irrigation (drought conditions) during the canopy establishment and under full irrigation is still necessary for a tropical savanna climate (Aw). This study is designed to determine the growth rate and starch yield of different cassava genotypes grown under irrigation and drought during canopy establishment.

## MATERIALS AND METHODS

### Experimental detail

This experiment was conducted under field conditions from August 2021 to August 2022 (2021/2022) and from August 2022 to August 2023 (2022/2023) at the Field Crop Research Station of Khon Kaen University, Khon Kaen, Thailand (16°28′ N, 102°48′ E, 200 m a.s.l.). The soil type for the experimental field was Yasothon Series (Yt: Oxic Paleustults). The experiment was a 2 × 6 split-plot design with four replications (main plot factor = water regime, subplot factor = genotype). Two water regimes, including drought conditions in the dry season and full irrigation, were assigned as main plots. Six cassava genotypes, Kasetsart 50, Rayong 9, Rayong 72, CMR38-125-77, CMR 35-91-63, and CM523-7, were assigned as subplots. The cassava genotypes were selected for high environmental adaptability (Kasetsart 50), high yield and high starch content (Rayong 9 and CMR38-125-77), high yield and drought tolerance (Rayong 72), high yield (CMR35-91-63), and low yield and drought tolerance (CM523-7).

Land preparation and tillage were conducted, and soil ridges were created with a 1 m distance between the ridges. The plot size was 7 × 10 m. Cassava stem cuttings of 20 cm from healthy 12-month-old plants were planted at 1 × 1 m spacing after soaking for 15 min in thiamethoxam (3-(2-chloro-thiazol-5-ylmethyl)-5-methyl-(1,3,5)-oxadia-zinan-4-ylidene-N-nitroamine, 25% water-dispersible granules) to prevent pest infestation. The stakes were inserted vertically to a depth of 14 cm into the soil ridges. Manual weed control was conducted between 30 to 90 days after planting (DAP). At 30 DAP, chemical fertilizer was applied according to the nutrient requirements for cassava as suggested by *Howeler (2002)* and the soil characteristics that were identified before planting. Chemical fertilizer (N-P-K) formula of 15-7-18 was applied at a rate of 312.5 kg ha$^{-1}$ at 60 DAP (*Department of Agriculture, 2010*). Before planting, soil at 0–30 cm and 30–60 cm depths were sampled to assess physical and chemical properties (Table 1). The soil texture at the Field Crop Research Station of Khon Kaen University was a sandy loam; the values for soil pH ranged from 6.3 to 6.8, total nitrogen varied from 0.2 to 0.3 g kg$^{-1}$, available phosphorus was between 8.0 to 36.8 mg kg$^{-1}$, and exchangeable potassium varied from 13.6 to 54.7 mg kg$^{-1}$. The soil chemical analysis indicated low total nitrogen and exchangeable potassium. From 30 to 90 days after planting (DAP), full irrigation based on a mini-sprinkler system
**Table 1  Physical and chemical properties of the soil for depths of 0–30 and 30–60 cm.**

| Soil property | 2021/2022 | | 2022/2023 | |
| --- | --- | --- | --- | --- |
| | 0–30 cm | 30–60 cm | 0–30 cm | 30–60 cm |
| Physical property | | | | |
| Texture class | Sandy loam | Sandy loam | Sandy loam | Loamy sand |
| Sand (%) | 75.0 | 71.0 | 82.9 | 67.9 |
| Silt (%) | 18.0 | 17.0 | 11.8 | 18.1 |
| Clay (%) | 7.0 | 12.1 | 5.4 | 14.1 |
| Chemical property | | | | |
| pH (1:1 $H_2O$) | 6.4 | 6.3 | 6.4 | 6.8 |
| Cation exchange capacity (cmol $kg^{-1}$) | 3.3 | 3.6 | 2.9 | 4.3 |
| Electrical conductivity (dS $m^{-1}$) | 0.03 | 0.02 | 0.02 | 0.02 |
| Organic matter (g $kg^{-1}$) | 4.3 | 2.9 | 2.9 | 1.9 |
| Total nitrogen (g $kg^{-1}$) | 0.3 | 0.2 | 0.2 | 0.2 |
| Available phosphorus (mg $kg^{-1}$) | 36.8 | 27.8 | 27.5 | 8.0 |
| Exchangeable potassium (mg $kg^{-1}$) | 54.7 | 21.3 | 13.6 | 18.2 |

was applied to all experimental plots under both irrigation and drought conditions. In the dry season (90 to 150 DAP), drought treatment was imposed by withholding irrigation, and supplementary irrigation was applied back for the recovery period from 151 to 360 DAP. For the plots that received full irrigation, the plants were irrigated throughout the crop duration. Irrigation was conducted based on the amount of crop water requirement (ETcrop) that was calculated as described by *Doorenbos & Pruitt (1992)*:

$$ET\ crop = ETo \times Kc \tag{1}$$

where ET crop is the crop requirement (mm $day^{-1}$), ETo is the evapotranspiration of a reference plant under specified conditions calculated by the pan evaporation method, and Kc is the crop water requirement coefficient that varies as a function of the growth stage. The Kc value for cassava was provided by the FAO, but it is inappropriate for the cassava growing conditions. The Kc value for FAO was calculated using a crop duration of 210 days. However, the crop duration of the cassava was 360 days. Therefore, we decided to use the Kc of sugarcane, which has a crop duration that covers 360 days (*Doorenbos et al., 1986*). In addition, the period for yield formation for cassava is also similar to sugarcane. The amount of water for irrigation was then calculated.

## Data collection

Soil moisture content was monitored by sampling at 90, 120, 150, 180, and 360 DAP for depths of 0–30 cm and 30–60 cm. The soil samples were oven-dried at 105 °C for 72 h or until weights were constant, and the moisture percentage was calculated. Soil moisture was determined by the gravimetric method described by *Shukla et al. (2014)* as shown below Eq. (2):

$$Soil\ moisture\ content\ (\%) = \frac{Soil\ wet\ weight\ (g) - Soil\ dry\ weight\ (g)}{Soil\ dry\ weight\ (g)} \times 100. \tag{2}$$

The relative water content (RWC) values were recorded at 180, 210, 240, 270, and 360 DAP. Measurements were taken from the second or third fully expanded leaves on the top of the main stem of two plants from each plot. The sampling occurred between 9:00 a.m. and 12:00 p.m. Leaf discs, each measuring one $cm^2$, were collected from 10 leaves per plot. Leaf fresh weight was determined in the laboratory and then soaked in distilled water for 24 h, after which it was weighed to obtain the turgid weight. Sampled leaves were oven-dried at 80 °C until constant weight, then weighed. RWC was calculated by following *Kramar (1980)*:

$$\text{RWC (\%)} = \frac{\text{Fresh weight (g)} - \text{Dry weight (g)}}{\text{Turgid weight (g)} - \text{Dry weight (g)}} \times 100. \tag{3}$$

Crop data was collected from two plants of each plot at 90, 120, 150, 180, and 360 DAP. The plants were separated into leaves, stems, storage roots, and fibrous roots. All plant parts were subsampled (about 10% of the total fresh weight of each organ). A subsample of fresh leaves was then used to measure leaf area by using a leaf area meter (LI-3100, LI-COR, Inc., Lincoln, NE, USA). Subsamples were oven-dried at 80 °C to achieve a constant dry weight. The harvest index (HI) was calculated as the ratio of the dry weight of storage roots to the total dry weight of the crop. The starch content of the storage root was assessed using the specific gravity method, and the starch yield was calculated by multiplying the starch content with the dry weight of the storage root. Calculations of CGR during 90 to 120 DAP, 120 to 150 DAP, 150 to 180 DAP, and 180 to 360 DAP were performed based on the function below (*Sawatraksa et al., 2019*) Eq. (3):

$$\text{CGR (gm}^{-2}\text{d}^{-1}) = (\frac{1}{G}) \times (\frac{DW_2 - DW_1}{T_2 - T_1}) \tag{4}$$

where G is sample area ($m^2$) and $DW_1$ and $DW_2$ are crop dry weight (g) at the times $T_1$ and $T_2(d)$. The equation for CGR was applied to calculate LGR, SGR, and SRGR.

Relative growth rate (RGR) was calculated using the following equation (*Sawatraksa et al., 2019*) Eq. (4):

$$\text{RGR (g g}^{-1}\text{ d}^{-1}) = \frac{\text{In }(DW_2) - \text{In }(DW_1)}{T_2 - T_1} \tag{5}$$

where $DW_1$ and $DW_2$ are crop dry weight (g) at the times $T_1$ and $T_2(d)$.

### Statistical analysis

Analysis of variance (ANOVA) was performed for all crop traits by following a model for split-plot design (*Gomez & Gomez, 1984*) and by using the statistix10 program (*Statistix10, 2013*). Mean comparisons were conducted for the least significant difference test (LSD) at $p \leq 0.05$.

## RESULTS

### Weather conditions

The weather data during the experimental periods at the Field Crop Research Station of Khon Kaen University shows that the drought treatment that was applied from November

to January coincided with the low rainfall period for both the 2021/2022 and 2022/2023 growing seasons (Fig. 1), which is generally observed for most of the years in Thailand. This period was recognized for its cooler and drier conditions, characterized by lower temperatures and reduced rainfall compared to other times. Average temperatures were slightly higher in the 2022/2023 growing season (22.2–31.6 °C) compared to the 2021/2022 season (21.6–30.5 °C). The daily solar radiation ranged from 5.7 to 26.4 MJ m$^{-2}$ in the 2021/2022 growing season and from 22.2 to 31.6 MJ m$^{-2}$ in the 2022/2023 growing season. The total rainfall was 1,957.8 mm in the 2021/2022 growing season and 1,592.7 mm in the 2022/2023 growing season.

## Soil moisture content

The information on soil moisture content during growing seasons was shown in Fig. 2, indicating that the values of soil moisture content for full irrigation treatment for the 2021/2022 growing season were close to field capacity (FC, 11.00% for 0–30 cm and 11.09% for 30–60 cm) (Figs. 2A, 2B). On the other hand, the soil moisture content in the drought treatment was lower than the FC values during canopy establishment (120–150 DAP). Specifically at 120 DAP, the soil moisture content was measured at 6.19% for the 0–30 cm depth and 8.73% for the 30–60 cm depth. The value of soil moisture content at a depth of 0–30 cm for 150 DAP (3.96%) was close to the permanent wilting point (PWP, 3.77%) (Fig. 2A). For the 2022/2023 growing season (Figs. 2C, 2D), the soil moisture contents at 120 DAP for the depths of 0–30 and 30–60 were 6.86% and 9.42%, respectively, lower than the FC values (11.03% for 0–30 cm and 11.02% for 30–60 cm). The soil moisture at 150 DAP reached a low of 4.39% at the 0–30 cm depth, indicating it was near the PWP value of 3.72% (Fig. 2C).

## Relative water content under different water regimes

The relative water content value (RWC) indicates the water content in a leaf at the time of sampling relative to its maximum water-holding capacity. The results revealed the different responses of six cassava genotypes in two water regimes. In comparison between water treatments, the drought treatment during the 2021/2022 and 2022/2023 growing seasons exhibited lower RWC values (89.81 and 92.59%, respectively) than the irrigated treatment (92.33 and 93.57%, respectively) (Fig. 3).

Based on the effect of the drought treatment during 90 to 150 DAP, the highest RWC values at 120 DAP were observed for Kasetsart 50 (89.95%), CMR38-125-77 (89.79%), and CM523-7 (89.50%) for 2021/2022 growing season and Rayong 9 (95.41%), Rayong 72 (94.76%), CMR38-125-77 (94.76%), and CM523-7 (94.10%) for 2022/2023 growing season, showed (Fig. 4). For 150 DAP, it marks the peak of a dry period, as evidenced by the very low soil moisture content (Fig. 2). The highest RWC values were observed from CMR35-91-63 for 2021/2022 growing season (96.72%) and Kasetsart 50 and CMR35-91-63 for 2022/2023 growing season (94.82 and 93.80%, respectively).

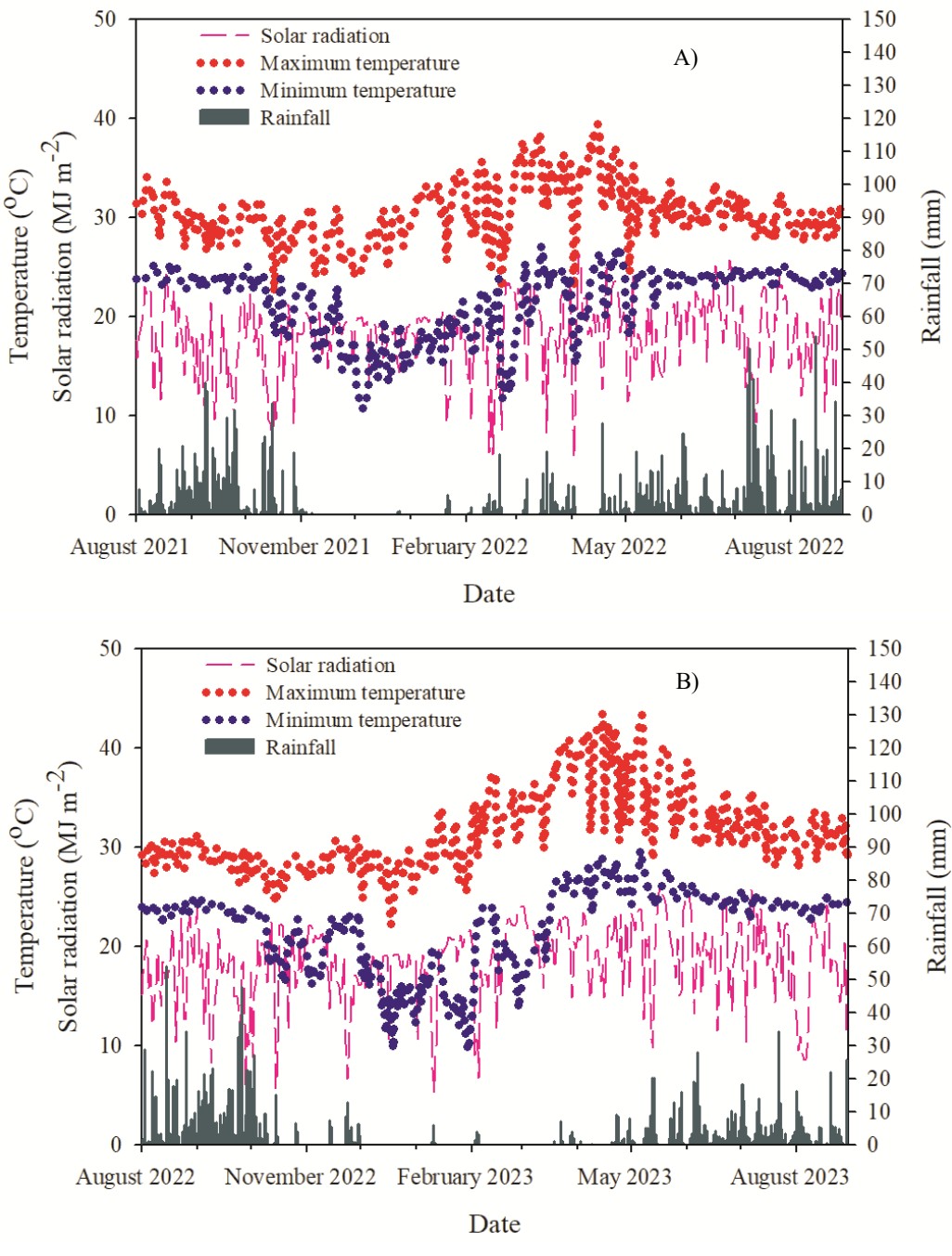

**Figure 1  Weather data at the Field Crop Research Station of Khon Kaen University, Khon Kaen, Thailand for the experiment from August 2021 to August 2022 and from August 2022 to August 2023.** (A) 2021/2022 and (B) 2022/2023. The daily weather data for crop duration.

## Evaluation of growth rates and performances of cassava at final harvest

In terms of growth rate for the 2021/2022 growing season, the results indicated the different responses of six cassava genotypes in two water regimes (Table 2). The drought treatment

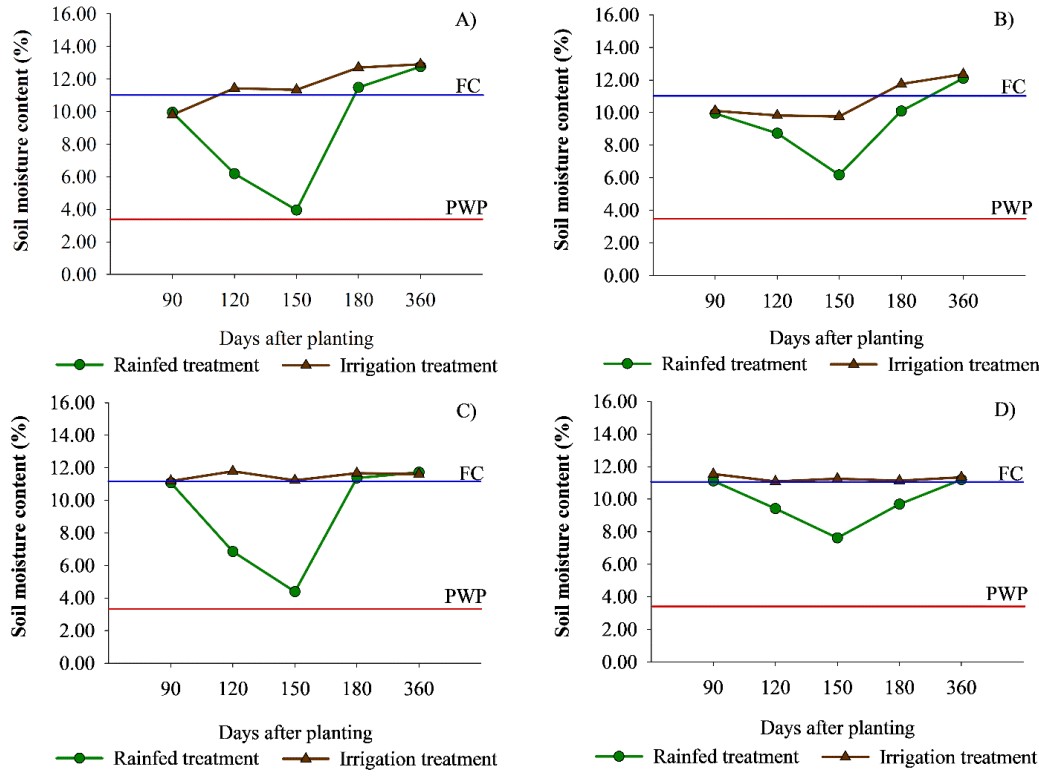

**Figure 2 Soil moisture content of rainfed treatment (drought treatment) and irrigation treatment.**
FC and PWP represent field capacity and permanent wilting point, respectively. (A) soil depth 0–30 cm in
2021/2022. (B) 30–60 cm in 2021/2022. (C) 0–30 cm in 2022/2023. (D) 30–60 cm in 2022/2023. Each data
point indicates the average value from experimental plots.

during 90 to 150 DAP provided lower SRGR (8.6 g m$^{-2}$ day$^{-1}$) than the irrigation treatment
(11.6 g m$^{-2}$ day$^{-1}$), but not for LGR and SGR. Comparing growth rates from 90 to 150
DAP among different cassava genotypes, Rayong 72 had the highest value of LGR (0.26 g
m$^{-2}$ day$^{-1}$), while CMR38-125-77 exhibited SGR (3.2 g m$^{-2}$ day$^{-1}$) and SRGR (15.4 g
m$^{-2}$ day$^{-1}$) that were higher than those of the other genotypes. After the drought during
the early growth phase, full irrigation was applied to all experimental plots, resulting in
higher LGR and SRGR (from 180 to 360 DAP) for the drought treatment (2.90 and 11.1 g
m$^{-2}$ day$^{-1}$, respectively) compared to the irrigation treatment (1.11 and 4.9 g m$^{-2}$ day$^{-1}$,
respectively). Among different cassava genotypes, the highest growth rate from 150 to 180
DAP were observed for CMR35-91-63 in terms of LGR and SGR (1.29 and 12.2 g m$^{-2}$
day$^{-1}$, respectively), and for Rayong 72 based on SRGR (15.5 g m$^{-2}$ day$^{-1}$). Kasetsart 50
displayed higher LGR, SGR, and SRGR from 180 to 360 DAP (3.06, 3.2, and 10.7 g m$^{-2}$
day$^{-1}$, respectively) compared to the other genotypes.

For CGR and RGR for the 2021/2022 growing season (Table 3), there were no significant
differences in these two parameters between the two water regimes during the 90 to 150
DAP. However, it was not the same among the two water regimes for CGR during 150 to
180 and 180 to 360 DAP, and RGR from 150 to 360 DAP. CMR38-125-77 exhibited the

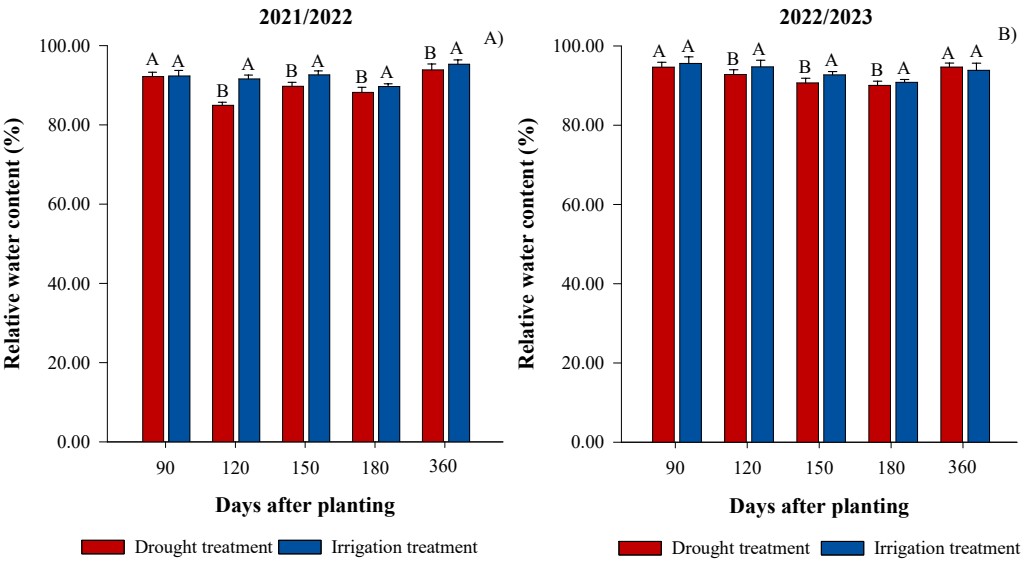

**Figure 3** **Relative water content (%) at 90, 120, 150, 180, and 360 days after planting (DAP) for drought and irrigation treatment.** (A) during 2021/2022 and (B) 2022/2023. Different letters in the same days after planting represent significant differences. Each data point indicates the average value from all cassava genotypes.

highest CGR values for all ranges: 90 to 150 DAP (19.7 g m$^{-2}$ day$^{-1}$), 150 to 180 DAP (32.6 g m$^{-2}$ day$^{-1}$), and 180 to 360 DAP (8.5 g m$^{-2}$ day$^{-1}$), compared to the other tested genotypes. Rayong 9 and CMR35-91-63 had a greater value of RGR for 150 to 360 DAP (0.62 and 0.64 g g$^{-1}$ day$^{-1}$, respectively) than the other genotypes. The different effects of two water regimes on six cassava genotypes were found, as indicated by an interaction between water treatment and genotype, except for RGR for 90 to 150 DAP.

According to the final harvest data for the 2021/2022 growing season (Table 4), the interaction between water regime and genotype for storage root fresh weight, storage root dry weight, total dry weight, HI, and starch yield indicated the various responses of six cassava genotypes in two water regimes. The drought treatment produced more storage root fresh weight (26.1 t ha$^{-1}$), total dry weight (12.8 t ha$^{-1}$), HI (0.77), and starch yield (2.50 t ha$^{-1}$) than the irrigation treatment. Comparing among genotypes, Rayong 72 and CMR38-125-77 performed well for almost all traits, except for total dry weight.

Based on the growth rate for the 2022/2023 growing season (Table 5), the drought treatment exhibited a significantly lower SGR during 90 to 150 DAP (0.3 g m$^{-2}$ day$^{-1}$) than the irrigation treatment (0.6 g m$^{-2}$ day$^{-1}$), but not for LGR and SRGR during the same period. Among the tested genotypes during 90 to 150 DAP, the highest growth rate values were observed for CMR38-125-77 regarding LGR (1.63 g m$^{-2}$ day$^{-1}$), Rayong 72 and CM523-7 for SGR (0.8 and 0.7 g m$^{-2}$ day$^{-1}$, respectively), and Rayong 72 for SRGR (2.5 g m$^{-2}$ day$^{-1}$). During the late growth phase, all experimental plots received full irrigation. This led to drought treatment having a higher LGR during 180 to 360 DAP and SRGR from 150 to 180 DAP (0.21 and 4.4 g m$^{-2}$ day$^{-1}$, respectively) compared to

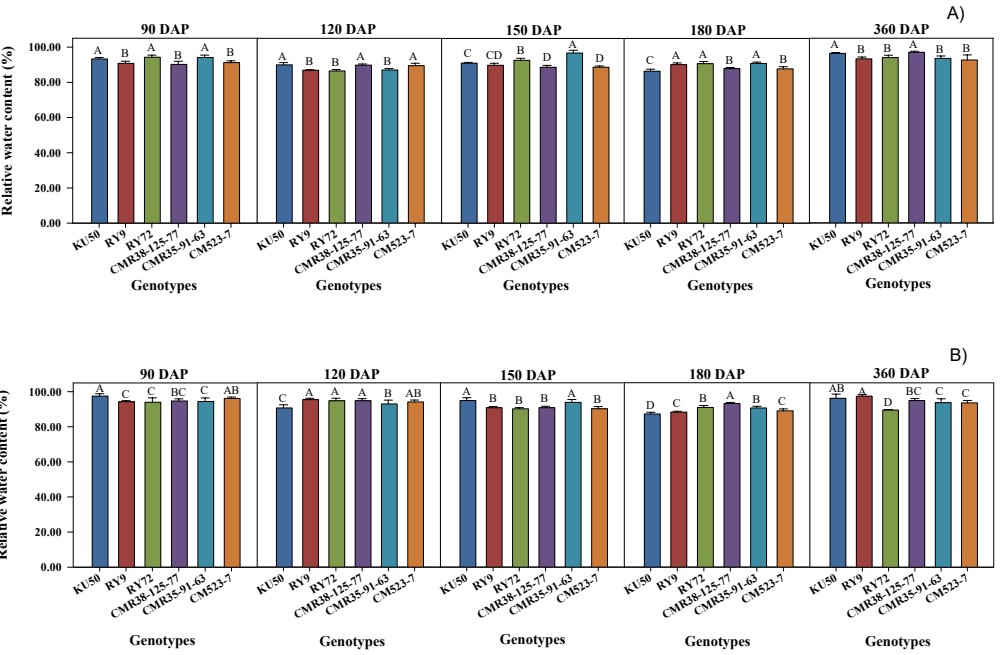

**Figure 4  Relative water content (%) at 90, 120, 150, 180, and 360 days after planting (DAP) for six cassava genotypes.** (A) during 2021/2022 and (B) 2022/2023. Different letters in the same days after planting represent significant differences. Each data point shows the average value for a cassava genotype.

the irrigation treatment (0.13 and 2.6 g m$^{-2}$ day$^{-1}$, respectively). Comparing among genotypes, the highest LGR values were recorded for CMR35-91-63 during 150 to 180 DAP (0.95 g m$^{-2}$ day$^{-1}$), and Rayong 72 from 180 to 360 DAP (0.23 g m$^{-2}$ day$^{-1}$). For SGR, Kasetsart 50, CMR38-125-77, and CMR35-91-63 had the highest values for 150 to 180 DAP (6.9, 6.6, and 6.7 g m$^{-2}$ day$^{-1}$, respectively). In a subsequent period, for 180 to 360 DAP, CMR35-91-63 maintained the highest SGR (0.7 g m$^{-2}$ day$^{-1}$). The highest SRGR for the ranges of 150 to 180 and 180 to 360 DAP was identified from CMR35-91-63 (6.3 and 7.0 g m$^{-2}$ day$^{-1}$, respectively). The results also showed an interaction between water regimes and cassava genotypes for LGR, SGR, and SRGR across all periods.

Regarding CGR and RGR for the 2022/2023 growing season (Table 6), the interaction between water regime and genotype indicated the response variation of six cassava genotypes to different water regimes. Among water regimes, CGR values during 90 to 150 and 150 to 180 for drought treatment (1.7 and 31.0 g m$^{-2}$ day$^{-1}$, respectively) were significantly smaller than for irrigation treatment (2.8 and 38.3 g m$^{-2}$ day$^{-1}$, respectively), but not for 180 to 360 DAP. A greater value of RGR from 150 to 360 DAP was recorded for the drought treatment (0.87 g g$^{-1}$ day$^{-1}$). In comparison between cassava genotypes, CMR35-91-63 showed the highest CGR from 90 to 150 (4.4 g m$^{-2}$ day$^{-1}$) and from 150 to 180 DAP (71.7 g m$^{-2}$ day$^{-1}$), and Rayong 9 recorded the highest CGR from 180 to 360 DAP (7.1 g m$^{-2}$ day$^{-1}$). Rayong 72 and CM523-7 demonstrated the highest RGR for 150 to 360 (0.89 and 0.87 g g$^{-1}$ day$^{-1}$, respectively).
**Table 2 Means for leaf growth rate (LGR), stem growth rate (SGR), and storage root growth rate (SRGR) during 90–150, 150–180, and 180–360 days after planting (DAP) of six cassava genotypes under two water regimes in the 2021/2022 growing season.** Each data point indicates the average for water treatment, genotype, and water treatment × genotype in the 2021/2022 growing season.

| Treatment | LGR (g m$^{-2}$ day$^{-1}$) | | | SGR (g m$^{-2}$ day$^{-1}$) | | | SRGR (g m$^{-2}$ day$^{-1}$) | | |
|---|---|---|---|---|---|---|---|---|---|
| | 90–150 DAP | 150–180 DAP | 180–360 DAP | 90–150 DAP | 150–180 DAP | 180–360 DAP | 90–150 DAP | 150–180 DAP | 180–360 DAP |
| Water treatment (W) | | | | | | | | | |
| Drought (W1) | 0.18A | 0.59B | 2.90A | 3.0A | 6.0B | 2.0 | 8.6B | 11.7 | 11.1A |
| Irrigation (W2) | 0.13B | 0.91A | 1.11B | 2.2B | 8.4A | 2.0 | 11.6A | 12.0 | 4.9B |
| F-test | * | * | ** | ** | ** | NS | ** | NS | ** |
| C.V. (%) | 29.53 | 16.74 | 28.14 | 15.29 | 2.94 | 20.85 | 7.92 | 8.97 | 21.45 |
| Genotype (G) | | | | | | | | | |
| Kasetsart 50 (G1) | 0.07E | 0.72B | 3.06A | 2.5B | 8.0B | 3.2A | 7.1E | 12.5B | 10.7A |
| Rayong 9 (G2) | 0.08E | 0.63B | 2.30B | 2.5B | 6.0D | 1.3C | 8.8D | 11.6B | 10.0A |
| Rayong 72 (G3) | 0.26A | 0.54B | 1.59C | 2.0C | 5.0E | 1.3C | 10.0C | 15.5A | 5.8C |
| CMR38-125-77 (G4) | 0.17C | 0.69B | 2.10B | 3.2A | 6.7C | 1.6C | 15.4A | 9.8C | 4.7C |
| CMR35-91-63 (G5) | 0.21B | 1.29A | 1.50C | 2.6B | 12.2A | 2.1B | 6.5E | 9.8C | 7.3B |
| CM523-7 (G6) | 0.14D | 0.65B | 1.51C | 2.7B | 5.6D | 2.5B | 12.8B | 12.0B | 9.56 |
| F-test | ** | ** | ** | ** | ** | ** | ** | ** | ** |
| G × W | | | | | | | | | |
| W1 × G1 | 0.04E | 0.55B-E | 4.70A | 3.5B | 6.8D | 3.7A | 5.8E | 7.0GH | 14.5A |
| W1 × G2 | 0.13D | 0.64B-E | 3.03C | 3.1BC | 7.1CD | 0.3E | 7.2DE | 8.2FG | 12.2B |
| W1 × G3 | 0.32A | 0.30E | 2.34D | 1.1FG | 4.9E | 1.1CD | 12.2B | 19.9A | 9.3C |
| W1 × G4 | 0.15CD | 0.87BC | 4.05B | 3.4B | 4.9E | 0.8DE | 13.1B | 13.6BC | 4.8EF |
| W1 × G5 | 0.30A | 0.80BC | 1.75E | 4.3A | 7.7C | 3.6A | 5.8E | 12.4CD | 13.1AB |
| W1 × G6 | 0.16B-D | 0.38DE | 1.50EF | 2.8CD | 4.8E | 2.8B | 7.3D | 9.3EF | 12.5AB |
| W2 × G1 | 0.11D | 0.89B | 1.41EF | 1.6EF | 9.1B | 2.8B | 8.4D | 18.0A | 7.0C-E |
| W2 × G2 | 0.04E | 0.63B-E | 1.57E | 1.9E | 4.9E | 2.3B | 10.4C | 15.0B | 7.8CD |
| W2 × G3 | 0.20B | 0.78B-D | 0.84F | 2.9B-D | 5.1E | 1.6C | 7.7D | 11.0DE | 2.2G |
| W2 × G4 | 0.20BC | 0.51C-E | 0.15G | 3.1BC | 8.4B | 2.5B | 17.7A | 5.9H | 4.5F |
| W2 × G5 | 0.12D | 1.77A | 1.19EF | 1.0G | 16.8A | 0.7DE | 7.2DE | 7.2GH | 1.5G |
| W2 × G6 | 0.11D | 0.92B | 1.51E | 2.5D | 6.4D | 2.2B | 18.2A | 14.7B | 6.6D-F |
| F-test | ** | ** | ** | ** | ** | ** | ** | ** | ** |
| C.V. (%) | 14.74 | 11.87 | 19.87 | 13.13 | 7.51 | 21.69 | 9.95 | 10.43 | 18.50 |

**Notes.**

Different letters in the same column represent significant differences (least significant difference test at $p \leq 0.05$). NS, non-significant.

*Significant at $p \leq 0.05$ level.

**Significant at $p \leq 0.01$ level.

In the final harvest data for the 2022/2023 growing season (Table 7), the responses of six cassava genotypes under two water regimes were different for all crop traits. The drought treatment had higher average values for storage root fresh weight, storage root dry weight, total crop dry weight, HI, and starch yield than the irrigation treatment. CMR38-125-77 is a desirable genotype for almost all crop traits, except for storage root fresh weight.

**Table 3   Means for crop growth rate (CGR) and relative growth rate (RGR) of six cassava genotypes under two water treatments in the 2021/2022 growing season.** Each data point indicates the average for water treatment, genotype, and water treatment × genotype in the 2021/2022 growing season.

| Treatment | CGR (g m$^{-2}$day$^{-1}$) | | | RGR ×10$^{-2}$ (g g$^{-1}$ day$^{-1}$) | |
|---|---|---|---|---|---|
| | 90–150 DAP | 150–180 DAP | 180–360 DAP | 90–150 DAP | 150–360 DAP |
| Water treatment (W) | | | | | |
| Drought (W1) | 14.7 | 18.1B | 9.1A | 2.2 | 0.49A |
| Irrigation (W2) | 15.3 | 40.6A | 3.6B | 2.7 | 0.39B |
| F-test | NS | ** | ** | NS | * |
| C.V. (%) | 5.25 | 6.75 | 13.04 | 30.60 | 16.77 |
| Genotype (G) | | | | | |
| Kasetsart 50 (G1) | 11.4D | 32.4AB | 6.4B | 2.0B | 0.41B |
| Rayong 9 (G2) | 17.8B | 18.0D | 8.1A | 2.8A | 0.62A |
| Rayong 72 (G3) | 18.4B | 28.4C | 5.6BC | 2.6A | 0.28D |
| CMR38-125-77 (G4) | 19.7A | 32.6AB | 8.5A | 2.4AB | 0.34C |
| CMR35-91-63 (G5) | 13.0C | 35.3A | 5.1BC | 2.1B | 0.64A |
| CM523-7 (G6) | 9.6E | 29.5BC | 4.5C | 2.8A | 0.34C |
| F-test | ** | ** | ** | ** | ** |
| G × W | | | | | |
| W1 × G1 | 10.1E | 39.2B | 5.8E-G | 2.0 | 0.45C |
| W1 × G2 | 10.6E | 2.8G | 11.1AB | 2.6 | 0.94A |
| W1 × G3 | 24.4A | 28.0D | 8.1CD | 2.6 | 0.26FG |
| W1 × G4 | 17.2C | 5.5G | 12.3A | 1.6 | 0.43CD |
| W1 × G5 | 12.4D | 14.0F | 9.6BC | 2.2 | 0.62B |
| W1 × G6 | 13.3D | 19.2E | 7.8C-E | 2.2 | 0.25FG |
| W2 × G1 | 12.7D | 25.6D | 6.9D-F | 2.1 | 0.36DE |
| W2 × G2 | 25.0A | 33.3C | 5.2F-H | 3.0 | 0.31EF |
| W2 × G3 | 12.3D | 28.7CD | 3.2HI | 2.6 | 0.31EF |
| W2 × G4 | 22.1B | 59.7A | 4.6GH | 3.2 | 0.24G |
| W2 × G5 | 13.5D | 56.7A | 0.7J | 2.1 | 0.67B |
| W2 × G6 | 5.8F | 39.8B | 1.11IJ | 3.4 | 0.42CD |
| F-test | ** | ** | ** | NS | ** |
| C.V. (%) | 6.35 | 11.15 | 11.12 | 18.35 | 8.82 |

**Notes.**

Different letters in the same column represent significant differences (least significant difference test at $p \leq 0.05$). NS, non-significant.

*Significant at $p \leq 0.05$ level.

**Significant at $p \leq 0.01$ level.

## DISCUSSION

This study focused on the growth analysis of different cassava genotypes under drought conditions during the canopy establishment and full irrigation. The findings can help select suitable cassava genotypes for dry periods during early growth and develop effective water management practices. The soil moisture content and RWC were used to explain water status in soil and crops, respectively, during the growing season. The RWC is a measure of the water status within the plant tissue (specifically the leaves), reflecting the water deficit experienced by the plant. Low rainfall decreased soil moisture contents and led to a low

**Table 4** **Means for storage root fresh weight, storage root dry weight, total dry weight, harvest index (HI), and starch yield at 360 days after planting (DAP) of six cassava genotypes under two water treatments in 2021/2022 growing season.** Each data point indicates the average for water treatment, genotype, and water treatment × genotype in the 2021/2022 growing season.

| Treatment | Storage root fresh weight (t ha$^{-1}$) | Storage root dry weight (t ha$^{-1}$) | Total dry weight (t ha$^{-1}$) | HI | Starch yield (t ha$^{-1}$) |
|---|---|---|---|---|---|
| Water treatment (W) | | | | | |
| Drought (W1) | 26.1A | 9.7 | 12.8A | 0.77A | 2.50A |
| Irrigation (W2) | 23.8B | 8.2 | 11.9B | 0.68B | 1.92B |
| F-test | * | NS | * | ** | ** |
| C.V. (%) | 9.13 | 19.72 | 7.04 | 2.36 | 9.52 |
| Genotype (G) | | | | | |
| Kasetsart 50 (G1) | 23.6B | 8.9BC | 13.4A | 0.67D | 2.49A |
| Rayong 9 (G2) | 23.2B | 8.5C | 11.3C | 0.71C | 2.12B |
| Rayong 72 (G3) | 25.7AB | 10.0A | 12.7B | 0.80A | 2.52A |
| CMR38-125-77 (G4) | 25.9AB | 9.7AB | 12.5B | 0.78AB | 2.60A |
| CMR35-91-63 (G5) | 23.7B | 8.3C | 11.1C | 0.75B | 2.23B |
| CM523-7 (G6) | 27.4A | 8.3C | 13.1AB | 0.64E | 1.28C |
| F-test | * | ** | ** | ** | ** |
| G × W | | | | | |
| W1 × G1 | 24.4B-D | 9.8B | 13.9BC | 0.73D | 2.96A |
| W1 × G2 | 26.3A-C | 9.8B | 11.1EF | 0.83A | 2.54B |
| W1 × G3 | 25.7A-C | 9.9B | 12.7D | 0.81A | 2.85A |
| W1 × G4 | 29.2A | 11.2A | 15.1A | 0.76B-D | 3.09A |
| W1 × G5 | 24.6B-D | 8.9BC | 11.9DE | 0.76B-D | 2.18C |
| W1 × G6 | 26.4A-C | 8.6B-D | 11.9DE | 0.73D | 1.35E |
| W2 × G1 | 22.8CD | 8.1CD | 12.9CD | 0.61E | 2.01C |
| W2 × G2 | 20.2D | 7.2D | 11.4E | 0.59EF | 1.71D |
| W2 × G3 | 25.8A-C | 10.0AB | 12.7D | 0.79A-C | 2.19C |
| W2 × G4 | 22.6CD | 8.2CD | 9.9G | 0.80AB | 2.11C |
| W2 × G5 | 22.8CD | 7.7CD | 10.2FG | 0.75CD | 2.28BC |
| W2 × G6 | 28.3AB | 8.0CD | 14.4AB | 0.56F | 1.20E |
| F-test | * | ** | ** | ** | ** |
| C.V. (%) | 11.73 | 8.85 | 5.46 | 4.09 | 8.92 |

**Notes.**

Different letters in the same column represent significant differences (least significant difference test at $p \leq 0.05$). NS, non-significant.

*Significant at $p \leq 0.05$ level.

**Significant at $p \leq 0.01$ level.

value of RWC (Figs. 1, 2 and 3). The relationship between RWC and soil moisture content was established, leading to the use of RWC values to identify suitable cassava genotypes across various water regimes in Thailand (*Ruangyos et al., 2024*; *Sawatraksa et al., 2018*; *Wongnoi et al., 2020*). The genotype with high RWC value during the dry periods serves as a mechanism for drought resistance, resulting from either enhanced osmotic regulation or reduced elasticity of tissue cell walls (*Ritchie, Nguyen & Holaday, 1990*). As indicated by high RWC values (Fig. 4) during the peak of the dry period (150 DAP) for both growing

**Table 5 Means for leaf growth rate (LGR), stem growth rate (SGR), and storage root growth rate (SRGR) during 90–150, 150–180, and 180–360 days after planting (DAP) of six cassava genotypes under two water regimes in the 2022/2023 growing season.** Each data point indicates the average for water treatment, genotype, and water treatment × genotype in the 2022/2023 growing season.

| Treatment | LGR (g m$^{-2}$ day$^{-1}$) | | | SGR (g m$^{-2}$day$^{-1}$) | | | SRGR (g m$^{-2}$day$^{-1}$) | | |
|---|---|---|---|---|---|---|---|---|---|
| | 90–150 DAP | 150–180 DAP | 180–360 DAP | 90–150 DAP | 150–180 DAP | 180–360 DAP | 90–150 DAP | 150–180 DAP | 180–360 DAP |
| Water treatment (W) | | | | | | | | | |
| Drought (W1) | 1.01A | 0.32B | 0.21A | 0.3B | 2.5B | 0.4B | 1.6A | 4.4A | 2.8B |
| Irrigation (W2) | 0.95B | 0.73A | 0.13B | 0.6A | 7.0A | 0.5A | 1.1B | 2.6B | 3.6A |
| F-test | * | ** | ** | ** | ** | ** | ** | ** | ** |
| C.V. (%) | 5.08 | 17.51 | 11.01 | 24.50 | 24.45 | 8.48 | 10.53 | 20.12 | 15.39 |
| Genotype (G) | | | | | | | | | |
| Kasetsart 50 (G1) | 1.32B | 0.50B | 0.07D | 0.2D | 6.9A | 0.4C | 0.8E | 4.4B | 2.1D |
| Rayong 9 (G2) | 0.64E | 0.30C | 0.21AB | 0.3C | 1.7D | 0.6B | 0.6E | 2.2D | 2.3CD |
| Rayong 72 (G3) | 0.79D | 0.33C | 0.23A | 0.8A | 2.7C | 0.6B | 2.5A | 3.7C | 1.3E |
| CMR38-125-77 (G4) | 1.63A | 0.60B | 0.16C | 0.5B | 6.6A | 0.2D | 2.1B | 2.1D | 3.6B |
| CMR35-91-63 (G5) | 0.99C | 0.95A | 0.15C | 0.3C | 6.7A | 0.7A | 1.0D | 6.3A | 7.0A |
| CM523-7 (G6) | 0.52F | 0.49B | 0.19B | 0.7A | 4.0B | 0.2D | 1.3C | 2.1D | 2.7C |
| F-test | ** | ** | ** | ** | ** | ** | ** | ** | ** |
| G × W | | | | | | | | | |
| W1 × G1 | 2.02B | 0.18E | 0.06EF | 0.2DE | 3.0E | 0.3E | 1.2D | 6.0BC | 1.6F |
| W1 × G2 | 0.24G | 0.45C | 0.26C | 0.2DE | 1.0FG | 0.7C | 0.6FG | 3.3D | 1.5F |
| W1 × G3 | 0.14G | 0.19E | 0.43A | 0.1G | 0.1G | 0.9B | 4.5A | 6.5B | 0.4G |
| W1 × G4 | 2.74A | 0.27DE | 0.04FG | 0.5C | 4.5CD | 0.1G | 1.2D | 1.3F | 4.0C |
| W1 × G5 | 0.50EF | 0.43CD | 0.14D | 0.1G | 3.5DE | 0.4E | 0.8EF | 7.6A | 7.5A |
| W1 × G6 | 0.45F | 0.43CD | 0.32B | 0.7B | 2.9E | 0.2F | 1.6C | 1.6EF | 1.6F |
| W2 × G1 | 0.63E | 0.81B | 0.08E | 0.1G | 10.8A | 0.5D | 0.4G | 2.8D | 2.6DE |
| W2 × G2 | 1.04D | 0.16E | 0.16D | 0.3D | 2.4EF | 0.5D | 0.7E-G | 1.2F | 3.1CD |
| W2 × G3 | 1.43C | 0.47C | 0.02G | 1.4A | 5.4C | 0.3E | 0.4G | 0.9F | 2.1EF |
| W2 × G4 | 0.50EF | 0.93B | 0.28C | 0.5C | 8.7B | 0.3E | 3.0B | 2.8D | 3.3CD |
| W2 × G5 | 1.49C | 1.47A | 0.16D | 0.4C | 9.8A | 1.1A | 1.3CD | 5.0C | 6.4B |
| W2 × G6 | 0.59EF | 0.54C | 0.07EF | 0.8B | 5.2C | 0.3E | 1.0DE | 2.6DE | 3.8C |
| F-test | ** | ** | ** | ** | ** | ** | ** | ** | ** |
| C.V. (%) | 11.57 | 21.29 | 10.46 | 16.48 | 14.75 | 7.83 | 12.77 | 15.22 | 14.53 |

**Notes.**

Different letters in the same column represent significant differences (least significant difference test at $p \leq 0.05$).

*Significant at $p \leq 0.05$ level.

**Significant at $p \leq 0.01$ level, respectively.

seasons (Fig. 2), CMR35-91-63 would be classified as a genotype with a good balance of the water content between leaves and water shortage conditions during the early growth phase.

The result revealed that even though cassava faces drought conditions during its early growth phase, some tested genotypes can still produce desirable results at final harvest if there is supplementary irrigation or rainfall in the later growth phase. The dry period from 90 to 150 DAP in this study, therefore, did not decrease the final yield for some tested genotypes, and ultimately produced slightly higher average values of biomass and yield
**Table 6 Means for crop growth rate (CGR) and relative growth rate (RGR) of six cassava genotypes under two water treatments in the 2022/2023 growing season.** Each data point indicates the average for water treatment, genotype, and water treatment × genotype in the 2022/2023 growing season.

| Treatment | CGR (g m$^{-2}$day$^{-1}$) | | | RGR × 10$^{-2}$ (g g$^{-1}$day$^{-1}$) | |
|---|---|---|---|---|---|
| | 90–150 DAP | 150–180 DAP | 180–360 DAP | 90–150 DAP | 150–360 DAP |
| Water treatment (W) | | | | | |
| Drought (W1) | 1.7B | 31.0B | 7.2A | 0.58 | 0.87A |
| Irrigation (W2) | 2.8A | 38.3A | 2.6B | 0.62 | 0.66B |
| F-test | ** | ** | ** | NS | ** |
| C.V. (%) | 7.42 | 0.50 | 3.43 | 7.74 | 11.13 |
| Genotype (G) | | | | | |
| Kasetsart 50 (G1) | 1.0E | 28.6C | 5.2C | 0.49C | 0.78B |
| Rayong 9 (G2) | 1.7D | 21.0D | 7.1A | 0.12E | 0.77BC |
| Rayong 72 (G3) | 1.0E | 17.0E | 6.5B | 0.35D | 0.89A |
| CMR38-125-77 (G4) | 3.6B | 42.0B | 3.5D | 0.48C | 0.71C |
| CMR35-91-63 (G5) | 4.4A | 71.7A | 0.7E | 1.41A | 0.57D |
| CM523-7 (G6) | 2.1C | 27.6C | 6.3B | 0.74B | 0.87A |
| F-test | ** | ** | ** | ** | ** |
| G × W | | | | | |
| W1 × G1 | 0.8F | 18.2G | 8.3C | 0.72D | 0.81C-E |
| W1 × G2 | 0.2G | 19.0G | 9.5B | 0.04H | 0.83B-D |
| W1 × G3 | 0.4G | 10.3H | 12.0A | 0.18G | 0.71F |
| W1 × G4 | 3.7B | 40.3D | 5.2E | 0.73D | 1.11A |
| W1 × G5 | 2.7D | 75.1A | 0.7G | 0.85C | 0.85BC |
| W1 × G6 | 2.5D | 23.0F | 7.3D | 0.96B | 0.90B |
| W2 × G1 | 1.1F | 39.0D | 2.2F | 0.25F | 0.72D-F |
| W2 × G2 | 3.1C | 22.9F | 4.6E | 0.20FG | 0.73D-F |
| W2 × G3 | 1.5E | 23.7F | 1.0G | 0.51E | 0.72EF |
| W2 × G4 | 3.4C | 43.7C | 1.8F | 0.23FG | 0.66F |
| W2 × G5 | 6.1A | 68.3B | 0.7G | 1.98A | 0.28G |
| W2 × G6 | 1.7E | 32.2E | 5.2E | 0.52E | 0.83BC |
| F-test | ** | ** | ** | ** | ** |
| C.V. (%) | 8.77 | 3.46 | 10.34 | 6.65 | 7.76 |

**Notes.**

Different letters in the same column represent significant differences (least significant difference test at $p \leq 0.05$). NS, non-significant.

[**]Significant at $p \leq 0.05$.

compared to the irrigation treatment (Tables 4 and 7). Cassava is a remarkably drought-resistant crop that can thrive with minimal water during its growth period (*El-Sharkawy, 1993*; *El-Sharkawy, De Tafur & Lopez, 2012*; *Howeler, 2002*; *Howeler, Lutaladio & Thomas, 2013*; *Sawatraksa et al., 2018*). *Santanoo et al. (2024)* conducted a single-year experiment on the photosynthetic performance and growth of different cassava genotypes grown under the dry period during the early growth phase and irrigation treatment. They found that net photosynthesis rate (Pn), petiole, root dry weight, leaf, stem, and storage root dry weight were reduced after 60 days of the dry period. After 30 days of re-watering, Pn fully recovered, leading to a significantly higher dry weight at 12 months after planting for the

**Table 7 Means for storage root fresh weight, storage root dry weight, total dry weight, harvest index (HI), and starch yield at 360 days after planting (DAP) of six cassava genotypes under two water treatments in the 2022/2023 growing season.** Each data point indicates the average for water treatment, genotype, and water treatment × genotype in the 2022/2023 growing season.

| Treatment | Storage root fresh weight (t ha$^{-1}$) | Storage root dry weight (t ha$^{-1}$) | Total dry weight (t ha$^{-1}$) | HI | Starch yield (t ha$^{-1}$) |
|---|---|---|---|---|---|
| Water treatment (W) | | | | | |
| Drought (W1) | 26.3A | 9.8A | 13.3A | 0.76A | 2.65A |
| Irrigation (W2) | 24.1B | 6.9B | 10.2B | 0.68B | 1.76B |
| F-test | ** | ** | ** | * | ** |
| C.V. (%) | 2.88 | 7.14 | 6.24 | 8.57 | 5.63 |
| Genotype (G) | | | | | |
| Kasetsart 50 (G1) | 22.9D | 8.6BC | 11.7B | 0.78A | 2.49A |
| Rayong 9 (G2) | 20.9E | 8.0C | 11.7B | 0.71BC | 2.13B |
| Rayong 72 (G3) | 30.1A | 6.9D | 10.1C | 0.65D | 1.98C |
| CMR38-125-77 (G4) | 25.9C | 9.7A | 12.9A | 0.75AB | 2.45A |
| CMR35-91-63 (G5) | 23.4D | 8.7B | 12.3AB | 0.70C | 2.10B |
| CM523-7 (G6) | 27.9B | 8.4BC | 11.7B | 0.72BC | 2.07EC |
| F-test | ** | ** | ** | ** | ** |
| G × W | | | | | |
| W1 × G1 | 21.8D | 9.6BC | 12.8BC | 0.80A | 3.09A |
| W1 × G2 | 24.3C | 9.2CD | 13.4B | 0.76AB | 2.47C |
| W1 × G3 | 29.6A | 10.3B | 13.4B | 0.77AB | 2.80B |
| W1 × G4 | 30.4A | 11.4A | 14.9A | 0.77AB | 2.96A |
| W1 × G5 | 25.8B | 9.6BC | 13.3B | 0.73BC | 2.59C |
| W1 × G6 | 25.9B | 8.7DE | 11.9CD | 0.73BC | 1.98E |
| W2 × G1 | 24.1C | 7.6FG | 10.6EF | 0.75AB | 1.90EF |
| W2 × G2 | 17.5E | 6.7G | 10.0F | 0.67C | 1.80F |
| W2 × G3 | 30.5A | 3.5H | 6.8G | 0.53D | 1.15H |
| W2 × G4 | 21.5D | 7.9EF | 10.9D-F | 0.72BC | 1.93EF |
| W2 × G5 | 21.0D | 7.7F | 11.4DE | 0.67C | 1.61G |
| W2 × G6 | 29.8A | 8.1EF | 11.4DE | 0.71BC | 2.17D |
| F-test | ** | ** | ** | ** | ** |
| C.V. (%) | 3.41 | 7.41 | 6.61 | 5.65 | 4.66 |

**Notes.**

Different letters in the same column represent significant differences (least significant difference test at $p \leq 0.05$).

*Significant at $p \leq 0.05$ level.

**Significant at $p \leq 0.01$ level.

drought treatment than the irrigation treatment. *Mahakosee et al. (2019)* planted cassava genotype cv. Rayong 9 under drought and irrigated conditions in Thailand. They found that the drought treatment with a planting date during the early growth phase, which had a dry period, produced higher storage root fresh weight, storage root dry weight, and total crop dry weight than the irrigation treatment.

The study on growth rate during different growing periods, along with crop dry weights at the final harvest, offers valuable insights into growth habits and enhances the understanding of adaptability. Although drought (90 to 150 DAP) diminished SRGR for 2021/2022 growing season, and SGR and CGR for 2022/2023 growing season, this treatment

displayed slightly higher values of LGR when compared to the irrigation treatment for both growing seasons (Tables 2 and 5). This is due to efficient leaf production under water-limited conditions in certain cassava genotypes, such as Rayong 72, CMR35-91-63, and CMR38-125-77, whose leaves continue to grow well despite water shortages. However, a better growth rate of the stem and storage root for the irrigation treatment led to a higher CGR from 90 to 150 DAP compared to the drought treatment (Tables 3 and 6). The results of this study indicate that although cassava experiences low water availability during the early growth phase, it is capable of recovering well when water is supplied again during the storage root development phase. This is evidenced by the high RGR and CGR between 150 and 360 DAP (after re-watering) in the drought treatment, which led to greater storage root fresh weight, total dry weight, HI, and starch yield compared to the full irrigation treatment throughout the entire crop duration (Tables 4 and 7). CGR in the late growth period was identified as a physiological determinant of storage root dry weight for cassava grown under different nitrogen applications (*Phuntupan & Banterng, 2017*) and various environments (*Phoncharoen et al., 2019a*).

Based on the average performance among cassava genotypes, this study highlighted that Rayong 72 (Table 4) and CMR38-125-77 (Tables 4 and 7) excelled in storage root dry weight, HI, and starch yield. The performance of these two cassava genotypes is associated with the growth rates of plant organs. For example, in the 2021/2022 growing season, Rayong 72 exhibited high LGR from 90 to 150 DAP (Table 2). Enhanced leaf growth during canopy establishment enables the plant to produce more photosynthates, resulting in greater storage root accumulation (*El-Sharkawy, 1993*; *El-Sharkawy, De Tafur & Lopez, 2012*; *Santanoo et al., 2024*). Meanwhile, CMR38-125-77 demonstrated high SGR and SRGR during the 90 to 150 DAP range. To determine the relationship between the final harvest data and CGR, however, high values of CGR for CMR38-125-77 during the 2021/2022 growing season are associated with high storage root dry weight, HI, and starch yield (Tables 3, 4, 6, and 7). A previous report has shown that not only does a higher CGR during the formation of storage roots support greater growth and yield, but also that a high LGR during storage root formation and a strong SRGR in the early growth phase are essential factors for enhancing cassava production (*Phuntupan & Banterng, 2017*). *Phoncharoen et al. (2019a)* reported that CGR and SRGR during 300-360 DAP and LGR during 60-120 and 300-360 DAP were the components for the physiological determinants of storage root dry weights for cassava genotypes grown under different planting dates. A report by *Sawatraksa et al. (2019)* on cassava grown in various environments also highlighted that specific growth rates, such as SGR, SRGR, and CGR, significantly correlated with total biomass and storage root dry weight.

A comparison among the combination of six genotypes and two different water regimes showed that CMR38-125-77 under drought treatments performed well in terms of storage root fresh weight, storage root dry weight, total dry weight, and starch yield for both the 2021/2022 and 2022/2023 growing seasons as compared to the other genotypes (Tables 4 and 7). This final harvest data of CMR38-125-77 under drought treatment related to high CGR during 180 to 360 DAP in the 2021/2022 growing season (Table 3) and a large value of RGR from 150 to 360 DAP in the 2022/2023 growing season (Table 6). This suggests

that the high productivity of cassava can be attributed to either the rapid accumulation of biomass over a specified period (CGR) (*Phuntupan & Banterng, 2017*) or the plant's strong ability to recover after experiencing drought stress (RGR) (*Awal & Ikeda, 2002*; *Abid et al., 2016*; *Ruangyos et al., 2024*; *Vandegeer et al., 2013*).

A study about growth analysis of different cassava genotypes grown under different planting dates by *Phoncharoen et al. (2019a)* and *Phoncharoen et al. (2019b)* reported that CMR 38-125-77 is likely to be an optimal genotype relative to total crop dry weight and storage root dry weight at final harvest for almost all growing dates. The previous study has also recorded the desirable performance in chlorophyll fluorescence of a CMR 38-125-77 genotype grown after rice harvesting and under rain-fed upper paddy field conditions (*Sawatraksa et al., 2018*). *Wongnoi et al. (2020)* have mentioned the desirable physiology, growth, and yield characteristics of a genotype CMR 38-125-77 grown in upland fields under a dry environment during the maximum storage root accumulation phase. A study by *Ruangyos et al. (2024)* regarding the evaluation of the physiological performance of different cassava genotypes grown under different irrigation levels also found that a CMR 38-125-77 had a high net photosynthesis rate compared to other genotypes.

The drought is generally observed in Thailand, where it is predominantly characterized by a tropical savanna climate (Aw). Growing cassava with a late rainy season planting date leads to the chance of the crop experiencing drought during the early growth. This study demonstrated the impact of drought during the early growth phase on growth and storage root yield for different cassava genotypes. For some cassava genotypes, however, the positive effect of drought during the early growth phase indicates drought resilience. These genotypes can be useful material for future breeding programs and cultivation under drought conditions during the early growth phase. There is another growing season of cassava in Thailand. For the early rainy season planting date, the drought can occur during the period of peak storage root accumulation. This is also important for breeders and farmers to pay keen attention to cassava. Additional research regarding drought for this growing period might help overcome this situation. In addition, the length of drought is also a crucial factor influencing these two growth phases of cassava, and this effect varies year to year due to climate change. The finding can also be applied to other regions with similar weather conditions.

Selection of the superior cassava genotypes under different growing conditions based on only final yield is inefficient, and analysis of morpho-physiological traits can provide useful information (*Phoncharoen et al., 2019a*; *Phuntupan & Banterng, 2017*; *Sawatraksa et al., 2019*). This study offers a better understanding of how particular cassava genotypes perform under drought during canopy establishment, and it could support prescient decision-making in identifying suitable genotypes within a given environment. This finding reinforces the critical role of water management in cassava cultivation and emphasizes the necessity of selecting genotypes with robust performance under both stress and recovery conditions. Further physiological characterization and molecular analysis of top-performing genotypes could provide insights into adaptive mechanisms and support targeted breeding for climate-resilient cassava production systems.

## CONCLUSIONS

The drought treatment during the canopy establishment decreased RWC for both growing seasons, SRGR (90 to 150 DAP) for 2021/2022 growing season, and SGR and CGR (90 to 150 DAP) for 2022/2023 growing season. Re-watering after the drought period could enhance the growth rate of cassava and produce a higher final yield and biomass than irrigation treatments. The preferred genotypes for storage root dry weight, HI, and starch yield were Rayong 72 and CMR38-125-77 for the 2021/2022 growing season, and CMR38-125-77 for the 2022/2023 growing season. In addition, the best performance in the storage root yield was CMR38-125-77 grown under drought treatment, and this was related to high CGR during 180 to 360 DAP in the 2021/2022 growing season and high RGR from 150 to 360 DAP in the 2022/2023 growing season. The identified cassava genotypes from this study are valuable material for future plant breeding and cultivation, aiming to enhance productivity in areas experiencing dry conditions during canopy establishment. Additional study on the impact of drought during the key growth stage, particularly the period of peak storage root accumulation for cassava, is also a crucial issue.

## ACKNOWLEDGEMENTS

We utilized Grammarly, a free artificial intelligence (AI) writing assistant, to refine the language and improve the quality of our manuscript.

### Funding

This study was supported by Khon Kaen University, Thailand and the Royal Golden Jubilee Ph.D. Program (N41A661180). Assistance in conducting the work was also received from the Plant Breeding Research Center for Sustainable Agriculture, Khon Kaen University and from the National Science and Technology Development Agency (NSTDA). The funders had no role in study design, data collection and analysis, decision to publish, or preparation of the manuscript.

### Grant Disclosures

The following grant information was disclosed by the authors:
Khon Kaen University, Thailand and the Royal Golden Jubilee Ph.D. Program: N41A661180.
The Plant Breeding Research Center for Sustainable Agriculture, Khon Kaen University and the National Science and Technology Development Agency (NSTDA).

### Competing Interests

The authors declare there are no competing interests.

### Author Contributions

- Passamon Ittipong conceived and designed the experiments, performed the experiments, analyzed the data, prepared figures and/or tables, authored or reviewed drafts of the article, and approved the final draft.

- Poramate Banterng conceived and designed the experiments, performed the experiments, analyzed the data, prepared figures and/or tables, authored or reviewed drafts of the article, and approved the final draft.
- Nimitr Vorasoot conceived and designed the experiments, performed the experiments, prepared figures and/or tables, and approved the final draft.
- Sanun Jogloy conceived and designed the experiments, performed the experiments, prepared figures and/or tables, and approved the final draft.
- Piyada Theerakulpisut conceived and designed the experiments, performed the experiments, prepared figures and/or tables, authored or reviewed drafts of the article, and approved the final draft.
- Kochaphan Vongcharoen performed the experiments, prepared figures and/or tables, and approved the final draft.
- Supranee Santanoo conceived and designed the experiments, performed the experiments, prepared figures and/or tables, and approved the final draft.

## Data Availability

The raw measurements are available in the Supplemental File.

## Supplemental Information

Supplemental information for this article can be found online at http://dx.doi.org/10.7717/peerj.20440#supplemental-information.

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
