# Peer review of "Morpho-physiological traits and yield quality for cassava genotypes planted under drought during canopy establishment"

_PeerJ, doi:10.7717/peerj.20440_

## Round 0.1 · original submission · Major Revisions

· Academic Editor

Major Revisions

Dear Authors
The manuscript cannot be accepted for publication in its current form. It needs a major revision before publication. The authors are invited to revise the paper, considering all the suggestions made by the reviewers. Please note that the requested changes are required for publication.
With Thanks

**Language Note:** The review process has identified that the English language must be improved. PeerJ can provide language editing services - please contact us at [email protected] for pricing (be sure to provide your manuscript number and title). Alternatively, you should make your own arrangements to improve the language quality and provide details in your response letter. – PeerJ Staff

Reviewer 1 ·

Basic reporting

Strengths:

The manuscript employs a generally clear structure, following scientific norms with appropriately placed sections (Introduction, Methods, Results, Discussion).

The literature cited is mostly relevant and recent, and the background adequately contextualizes the research question.

Figures and tables are high quality, well-labelled, and integral to the understanding of the results.

Critical Observations:

While the English language is understandable, the manuscript would benefit substantially from professional editing. The sentence structures are often rigid and awkward, with transitions between ideas lacking fluidity. This hinders readability, especially for an international audience.

There are instances of overly technical phrasing that obscure rather than clarify meaning. Some key terms and abbreviations (e.g., RWC, SRGR) are introduced without sufficient explanation at first mention.

Several sentences are overly long or convoluted, particularly in the abstract and discussion, reducing the clarity of the scientific narrative.

Citation formatting is inconsistent throughout. Some references lack DOIs or present them without hyperlinks, and the formatting style appears to fluctuate between entries.

While the structure generally conforms to PeerJ standards, the overall tone and fluency would benefit from revision by a native or professional science editor to ensure the manuscript meets the linguistic expectations of high-impact international journals.

Recommendation: A comprehensive language revision by a professional editing service or a fluent English-speaking academic in the field is strongly recommended before publication.

Experimental design

Strengths:

The two-year field experiment using a 2 × 6 split-plot design provides robustness to the findings.

The selection of six cassava genotypes with varying drought tolerance enables meaningful genotype × environment interaction analysis.

Soil and climatic data are included, enhancing the reproducibility and environmental relevance of the results.

Critical Observations:

The Kc values used for calculating crop water requirement (ETcrop) are not linked to a specific or cited source. Although the formula from Doorenbos & Pruitt (1992) is mentioned, the Kc stage-specific coefficients are neither cited nor justified, compromising transparency.

Soil moisture measurement methodology is sound, but details on calibration or sensor accuracy are omitted.

Prior multi-location or historical performance data of genotypes are not referenced, which would have contextualized genotype selection.

At most, there are critical errors in statistical analyses. I have detailed their reasons in the attached file.

Validity of the findings

Growth rate indicators (CGR, RGR, LGR, SGR) were calculated for specific developmental stages, which adds resolution to temporal responses. Thus, ANOVA and LSD are not appropriate for analyzing split-plot interactions. Repeated measures are more suitable for this data.

Confidence intervals or standard errors are not consistently reported, which reduces the interpretability of statistical outcomes.

Additional comments

GENERAL COMMENTS
Positive Aspects:

The study contributes valuable data on genotype performance under drought stress, a critical area in climate-resilient crop research.

The inclusion of both early-stage stress and late-stage recovery phases allows for a nuanced understanding of cassava’s adaptive potential.

Genotypes Rayong 72 and CMR38-125-77 are convincingly demonstrated to outperform under stress conditions, reinforcing their breeding value.

Limitations:

The discussion section mainly reiterates results and lacks critical engagement with contradictory or ambiguous findings.

The practical implications for farmers, policy makers, or cassava breeding programs are underdeveloped.

The conclusion merely restates findings; it lacks strategic insight into broader impacts or future directions.

Annotated reviews are not available for download in order to protect the identity of reviewers who chose to remain anonymous.

Reviewer 2 ·

Basic reporting

No comment

Experimental design

No comment

Validity of the findings

no comment

Additional comments

This research article investigates the impact of drought during the canopy establishment phase on the growth and starch yield of six different cassava genotypes, revealing that a period of drought followed by rewatering can actually enhance yield and biomass compared to continuous irrigation, with genotypes like Rayong 72 and CMR38-125-77 showing particular promise for drought tolerance and high starch production, ultimately contributing valuable insights for selecting cassava varieties adaptable to water-stressed conditions.
-Comments and Suggestions for Authors
1- Line 41: Thailand is a major cassava producer that produced... could be more concisely stated as "Thailand is a major cassava producer, with an output of..."
2- Line 46: Cassava production in this region has been planted in two seasons, i.e., the rainy and the late rainy seasons" is slightly wordy. Consider: Cassava in this region is typically cultivated in two seasons: the rainy and late rainy seasons."
3- Line 53: Recommendation of the cassava genotypes that can adapt well under drought conditions is a strategy to help the farmer in achieving high productivity with low investment. Rephrase to: Recommending drought-adaptive cassava genotypes is a strategy to help farmers achieve high productivity with low investment.
4- Line 45: which has sandy soils with poor soil fertility, low soil water holding capacity, and unpredicted rainfall. While factually correct, consider adding a connecting word for better flow: "...the Northeast, characterized by sandy soils with poor soil fertility, low soil water holding capacity, and unpredictable rainfall."
5- Line 49: (the canopy establishment) - While the parenthesis provides context, integrating it more smoothly into the sentence might improve readability. For example: "...drought during the early growth phase, specifically canopy establishment..."
6- Line 69: ...early drought (Santanoo et al., 2024). Consider using "under early drought conditions" for consistency.
7- Line 70: ...and an additional experiment is necessary for better explanations. Could be more concise: "...necessitating further research for more robust conclusions."
8- Line 92: 200 m asl - Should be "200 m a.s.l." for "meters above sea level."
9- Line 106: 15-7-18 (N-P2O5-K2O) - While common, it's better to explicitly state that the phosphorus and potassium are in oxide forms upon first mention or consistently use elemental forms if that's the standard in the field.
10- Line 93: "The experiment was a 2 x 6 split plot design with four replications." This is clear, but briefly stating the factors for the split plot (main plot factor = water regime, subplot factor = genotype) could enhance understanding for readers unfamiliar with the specific experimental design.
11- Line 97-99: The descriptions of why specific genotypes were chosen are helpful but could be slightly more concise and consistently phrased. For example, instead of "identified as highly adapted to the environment (Kasetsart 50)," consider "selected for high environmental adaptability (Kasetsart 50)."
12- Line 102: "planted at 1 x 1 m" - Specify if this is plant spacing within rows and between rows, or if it's a square planting pattern.
13- Line 104: "2/3 length of the stick" - While understandable, providing an approximate depth in cm might be more scientifically precise.
14- Line 107: "From 30 to 90 days after planting (DAP), full irrigation was applied to all experimental plots." This seems contradictory to the main plot assignment of drought and full irrigation. It needs clarification. Was there an initial establishment phase with full irrigation for all plots before treatments were imposed? This is a crucial detail.
15- Line 108: "In the dry season (90 to 150 DAP), drought treatment was imposed..." It would be beneficial to briefly describe how the drought treatment was imposed (e.g., cessation of irrigation).
16- The discussion section, Line 239: "The RWC value was used to indirectly measure soil water status..." This statement is scientifically inaccurate. Relative Water Content (RWC) is a measure of the water status within the plant tissue (specifically the leaves), reflecting the water deficit experienced by the plant. It is not a direct or indirect measure of soil water status. Soil moisture content (as mentioned in line 236 and Figure 2) is the appropriate metric for assessing water availability in the soil. The authors might be implying a correlation between soil water status and plant RWC, but the current phrasing is misleading.
17- Line 251-252: "...did not decrease the final yield but produced more biomass and yield than the irrigation treatments (Tables 4 and 7)." This is a significant claim that contradicts the general understanding of plant physiology, where water stress typically leads to reduced growth and yield, especially during critical early growth stages. While some studies (cited later) suggest that early drought followed by rewatering can sometimes lead to compensatory growth, the statement as presented is a broad generalization that needs careful justification and consideration of potential mechanisms and limitations. It's crucial to examine Tables 4 and 7 to verify this claim and understand the magnitude of the difference and its statistical significance.
18- Line 266-269: "The drought treatment exhibited greater values of LGR than irrigation treatment during 90 to 150 DAP... This pointed out that not watering cassava during the dry season, when it was 90 to 150 DAP, had a positive effect on leaf growth." This is a counterintuitive finding. Drought stress generally inhibits leaf expansion and growth due to reduced turgor pressure and metabolic limitations. While specific genotypes might exhibit some level of osmotic adjustment or other mechanisms to maintain leaf function under mild stress, a greater leaf growth rate under drought compared to irrigation requires a very strong and specific explanation. The authors should elaborate on potential mechanisms for this observation, such as altered resource allocation or specific responses of the studied genotypes. Simply stating that not watering had a "positive effect" lacks mechanistic insight.
19- Line 275-277: "...cassava grown under the drought treatment during 90 to 150 DAP and receiving full irrigation during 151 to 360 DAP gave higher CGR from 180 to 360 DAP and RGR from 150 to 360 DAP, leading to more storage root dry weight, total dry weight, HI, and starch yield than the full irrigation treatment for the whole crop duration (Tables 4 and 7)." Similar to point 2, this claim of drought treatment leading to higher final yield components than full irrigation needs robust justification.

·

Basic reporting

The manuscript titlted about is a very well composed and conducted research by the authors with a great deal of value for cassava breeding not only in Thailand but also globally. However, serious improvements need to be made to improve the quality of the manuscript. These critiques will help to improve the manuscript and therefore need to be meticulously address.
 In the abstract,
• What are the main findings on the growth rate and starch yield?
• Provide the range of results on the growth rate and starch yield of the genotypes, and the averages of each treatment.
• Did the treatment cause any significant disparities in the results? Indicate that in the abstract.

All other critiques can be found on the pdf file.

Experimental design

Experiment was well designed

Validity of the findings

Areas of concern are hihhlighted in the pdf file

---

## Round 0.2 · Major Revisions

· Academic Editor

Major Revisions

Dear Authors

The manuscript cannot be accepted for publication in its current form. It still needs a major revision before publication. The authors are invited to revise the paper, taking into account all the suggestions made by the reviewers. Please note that the requested changes are required for publication.

With Thanks

**Language Note:** The review process has identified that the English language must be improved. PeerJ can provide language editing services - please contact us at [email protected] for pricing (be sure to provide your manuscript number and title). Alternatively, you should make your own arrangements to improve the language quality and provide details in your response letter. – PeerJ Staff

Reviewer 1 ·

Basic reporting

This is the second time for reviewing the manuscript. I rechecked the manuscript and it can be stated that authors carefully revised the manuscript according to the reviewer's comments. This version of the manuscript can be published.

Experimental design

Some critical errors and omissions were corrected by the authors. It looks better.

Validity of the findings

After revision of the statistical analysis section, the validity of the data increased

Reviewer 2 ·

Basic reporting

no comment

Experimental design

no comment

Validity of the findings

no comment

Additional comments

The authors have made the changes I suggested in the last review. I recommend its publication in this journal.

·

Basic reporting

Review report for “Morpho-physiological traits and yield quality for cassava genotypes planted under drought during canopy establishment (#116450)”

Decision: minor review

The authors presented significant findings for cassava farmers and breeders in this research, especially in water crisis ecosystems. However, a few critiques are presented below.

1. In the results section, authors should section it into subheadings. Suggestions can be found in the PDF file.

2. Which period within the study is the most critical for breeders and farmers to pay keen attention to? Authors should clarify that in the results, buttress it in the discussion and make a statement to conclude it in the conclusion.

All in all, this was well-constructed research with very significant findings.

Experimental design

All other suggestions can be found in the attached file.

Validity of the findings

All other suggestions can be found in the attached file.

---

## Round 0.3 · Minor Revisions

· Academic Editor

Minor Revisions

Dear Authors
The manuscript still needs a minor revision before publication. The authors are invited to revise the paper, considering all the suggestions made by the reviewer.
With Thanks

·

Basic reporting

No comment

Experimental design

No comment

Validity of the findings

The authors have made a significant effort to improve the manuscript, especially addressing my earlier critiques. However, in the results section, the range of values needs to be provided briefly before confirmation is made in the tables or figures. Authors should provide a brief range of values for each section, indicating the minimum, maximum, and average for each season, to facilitate easy comparison in the text.

Additional comments

The authors have made a significant effort to improve the manuscript, especially addressing my earlier critiques. However, in the results section, the range of values needs to be provided briefly before confirmation is made in the tables or figures. Authors should provide a brief range of values for each section, indicating the minimum, maximum, and average for each season, to facilitate easy comparison in the text.

---

## Round 0.4 · Minor Revisions

· Academic Editor

Minor Revisions

Dear Authors,
I am pleased to inform you that the manuscript has been improved following the last revision abd is almost ready to be accepted.

Firts, please address these items noted by the Secion Editor:

1) The authors acknowledge "Acknowledgments We utilized an artificial intelligence (AI) tool to edit our manuscript." However, this should go into "Declarations" and provide more detail about the used tools, and state the responsibility of the authors for the final manuscript.

2) The writing must be improved. Currently, the manuscript is difficult to understand and too verbose. Example (from the Conclusions): "The drought treatment during the canopy establishment decreased soil moisture contents, RWC, SGR, SRGR, and CGR (from 90 to 150 DAP). Re-watering after the drought period could enhance the growth rate of cassava and produce a higher final yield and biomass than irrigation treatments. ".. The drought reduced the soil moisture? This is actually the definition of drought, and not even a result. Please be more precise and integrate the results for the conclusions. My comments also apply for the rest of the text; e.g. the Abstract.

With Thanks

·

Basic reporting

No comment

Experimental design

No comment

Validity of the findings

No comment

Additional comments

The authors have incorporated my comments, which have improved the quality of the manuscript. There are no further comments.

---

## Round 0.5 · accepted · Accept

· Academic Editor

Accept

Dear Authors,

I am pleased to inform you that the manuscript can be accepted for publication.
Congratulations on accepting your manuscript, and thank you for your interest in submitting your work to PeerJ.

With Thanks